# Efficacy and Safety of Pulse Intravenous Methylprednisolone in Pediatric Epileptic Encephalopathies: Timing and Networks Consideration

**DOI:** 10.3390/jcm13092497

**Published:** 2024-04-24

**Authors:** Angelo Russo, Serena Mazzone, Laura Landolina, Roberta Colucci, Flavia Baccari, Anna Fetta, Antonella Boni, Duccio Maria Cordelli

**Affiliations:** 1IRCCS, Istituto delle Scienze Neurologiche di Bologna, UOC Neuropsichiatria Dell’età Pediatrica, 40139 Bologna, Italy; serena.mazzone@ausl.bologna.it (S.M.); laura.landolina@studio.unibo.it (L.L.); colucciroby@yahoo.it (R.C.); anna.fetta2@unibo.it (A.F.); antonella.boni@isnb.it (A.B.); ducciomaria.cordelli@unibo.it (D.M.C.); 2Dipartimento di Scienze Mediche e Chirurgiche (DIMEC), Università di Bologna, 40126 Bologna, Italy; 3IRCCS, Istituto delle Scienze Neurologiche di Bologna, UOS Epidemiologia e Biostatistica, 40139 Bologna, Italy; flavia.baccari@ausl.bologna.it

**Keywords:** methylprednisolone, pulse therapy, childhood epilepsy, epileptic encephalopathy

## Abstract

**Background:** Epileptic encephalopathies (EE) are characterized by severe drug-resistant seizures, early onset, and unfavorable developmental outcomes. This article discusses the use of intravenous methylprednisolone (IVMP) pulse therapy in pediatric patients with EE to evaluate its efficacy and tolerability. **Methods:** This is a retrospective study from 2020 to 2023. Inclusion criteria were ≤18 years at the time of IVMP pulse therapy and at least 6 months of follow-up. Efficacy and outcome, defined as seizure reduction > 50% (responder rate), were evaluated at 6 and 9 months of therapy, and 6 months after therapy suspension; quality of life (QoL) was also assessed. Variables predicting positive post-IVMP outcomes were identified using statistical analysis. **Results:** The study included 21 patients, with a responder rate of 85.7% at 6 and 9 months of therapy, and 80.9% at 6 months after therapy suspension. Variables significantly predicting favorable outcome were etiology (*p* = 0.0475) and epilepsy type (*p* = 0.0475), with the best outcome achieved in patients with genetic epilepsy and those with encephalopathy related to electrical status epilepticus during slow-wave sleep (ESES). All patients evidenced improvements in QoL at the last follow-up, with no relevant adverse events reported. **Conclusions**: Our study confirmed the efficacy and high tolerability of IVMP pulse therapy in pediatric patients with EE. Genetic epilepsy and ESES were positive predictors of a favorable clinical outcome. QOL, EEG tracing, and postural–motor development showed an improving trend as well. IVMP pulse therapy should be considered earlier in patients with EE.

## 1. Introduction

Refractory epilepsy poses a significant challenge, affecting approximately a quarter of children with epilepsy [1].

Among the various presentations of childhood epilepsy, epileptic encephalopathies (EE) represent an important subgroup characterized by severe and drug-resistant seizures, early onset, and unfavorable developmental outcomes [2].

According to the International League Against Epilepsy, “epileptic encephalopathies” are defined as conditions wherein “the epileptic activity itself may contribute to severe cognitive and behavioral impairments above and beyond what might be expected from the underlying pathology alone (e.g., cortical malformation), and that these can worsen over time” [2,3]. In certain instances, developmental slowing may precede the onset of frequent epileptic activity on EEG; in such cases the term “developmental and epileptic encephalopathies” is suggested [2,3].

Despite the introduction of new drugs with multiple targets in the past 30 years, EE continues to exhibit poor response to anti-seizure medications (ASMs) [4,5,6].

Hence, there is an urgent need to complement ASMs with other therapeutic modalities in these severe epileptic conditions, including vagus nerve stimulation (VNS), ketogenic diet and corticosteroid therapy [7,8,9].

Steroid therapy has been effectively utilized in drug-resistant epilepsy for many years [10,11,12,13,14,15,16,17,18,19,20,21,22,23,24,25,26,27,28], although its precise mechanism of action on brain activity remains incompletely understood [29,30,31,32,33,34,35,36,37,38,39,40].

Various steroid formulations are available for the treatment of EE, with oral prednisolone, adrenocorticotropic hormone (ACTH), and methylprednisolone being among the most commonly used options.

To mitigate potential adverse events associated with long-term oral steroid and short-term intramuscular ACTH therapy [41], pulse intravenous methylprednisolone (IVMP) is frequently employed in patients with EE. Different therapeutic strategies are reported, including varying dosages (ranging from 15 to 30 mg/kg/day), frequencies (such as one pulse per week or one pulse per month), and durations (spanning from 1 to 36 months) [19,20,21,22,23,24,25,26,27,28]. Additionally, in some cases, oral prednisone therapy followed an initial IVMP pulse [20,24].

This study presents findings from a single-center experience involving a cohort of pediatric patients diagnosed with epileptic encephalopathies who underwent intravenous methylprednisolone (IVMP) pulse therapy. The objectives were to assess its efficacy and tolerability, as well as to determine the minimum duration of therapy required to achieve optimal response and outcomes.

Additionally, we conducted an analysis to assess the contribution of factors predicting favorable post-IVMP outcomes.

## 2. Materials and Methods

We conducted a retrospective study, analyzing the clinical records of pediatric patients diagnosed with epileptic encephalopathy (EE), according to the ILAE classification [2], undergoing long-term intravenous methylprednisolone (IVMP (Pfizer Italia S.r.l., Latina, Italy)) at the IRCCS—Institute of Neurological Sciences of Bologna between 2020 and 2023.

All patients were collectively categorized as a single group of EE for study purposes, although separate analyses were also performed by dividing this category into patients suffering from encephalopathy related to electrical status epilepticus during slow wave sleep (ESES) and those suffering from other early-onset EE.

We included patients who were under 18 years of age at the time of steroid therapy and had a minimum follow-up period of 6 months after the last infusion.

To mitigate potential confounding effects from other therapeutic variables, we excluded patients with epileptic encephalopathy who were undergoing treatment with ketogenic diet or vagus nerve stimulation, and patients who needed changes in their chronic anti-seizure medications (ASMs) during IVMP treatment.

Each cycle of IVMP pulse therapy comprised administration of intravenous methylprednisolone at a dosage of 20 mg/kg/day for three consecutive days [13]. Multiple cycles of IVMP were administered at monthly intervals, with a total treatment duration of 9 months, as per our institutional protocol. Following pulse administration, a gradual tapering of steroids through oral cortisone was not carried out.

The response to therapy was evaluated based on seizure frequency. Accordingly, in post-IVMP data analysis, the responder rate was determined by a reduction in seizures of ≥50%. Patients with a response of less than 50% were defined as non-responders. Furthermore, patients were classified as partial responders when the seizure reduction was between 50% and 75% and good responders when the reduction was ≥75%. Seizure freedom (SF) was defined as the complete cessation of all seizures (seizure frequency = 0) at the time of follow-up.

The responder rate was assessed during the first 6 months and at 9 months of IVMP therapy. The outcome was analyzed 6 months after the completion of IVMP therapy. During the first 6 months of therapy, the month wherein results demonstrated at least a partial response in more than 50% of the population was considered as the minimum time to evaluate the effectiveness of the therapy in our clinical practice.

In addition to assessing the main response to IVMP therapy, which focused on seizure frequency, we also analyzed a secondary response, related to the quality of life (QoL) of patients and caregivers, EEG findings, and postural-motor development.

The clinical variables examined retrospectively included: age at epilepsy onset, epilepsy duration before the IVMP pulse therapy, seizure frequency, type of epilepsy, etiology, neurological examination findings, neuropsychological evaluation results, and MRI findings.

Pre-IVMP and post-IVMP seizure frequency data were retrospectively assessed by reviewing patient medical records. In our seizure unit’s clinical practice, accurate collection of clinical data, including seizure frequency and duration, is conducted monthly during steroid therapy. QoL data were also collected monthly from the initiation until the completion of the treatment. Data were obtained by querying caregivers about parameters (alertness, school achievement, mood, seizure intensity, postictal state) and the overall lifestyle of the patients. Additionally, a caregiver QoL parameter was included to account for the potential emotional fatigue accompanying the management of patients with EE. QoL results were designated as improved, unchanged, or worse, compared to the pre-IVMP therapy.

EEG findings and postural–motor development were also evaluated during the course of treatment and 6 months post-IVMP therapy.

The outcome was deemed favorable when there was a seizure reduction of more than 50% at the last follow-up.

Database analyses were conducted in accordance with institutionally approved human subject protection protocols.

### Statistical Analysis

We conducted statistical analysis to assess the associations between certain variables and the overall outcome. Continuous variables were presented as mean ± standard deviation (SD), while categorical variables were summarized using absolute and relative frequencies (%).

Fisher’s exact test and Wilcoxon rank-sum tests were utilized, as appropriate, to assess the univariate association between the overall outcome and each individual variable collected, including epilepsy onset, epilepsy duration before IVMP (timing), seizure frequency, age at of epilepsy, etiology, neurological examination findings, neuropsychological evaluation results, and MRI findings. All *p*-values were based on two-sided tests, with significance set at *p* < 0.05.

Multiple logistic regression analysis was also performed to assess the relationship between the outcome and multiple predictor variables simultaneously at each outcome point. The results are presented as an odds ratio (OR) along with its respective 95% confidence interval (CI). Statistical analysis was performed using the statistical package Stata SE, version 14.0.

## 3. Results

### 3.1. Patients

The cohort comprised 21 children, 12 males and 9 females. The mean age at beginning of intravenous methylprednisolone (IVMP) therapy was 92.9 ± 38.3 months, with a mean age at seizure onset of 27.4 ± 25.8 months and a pretreatment epilepsy duration of 65.5 ± 33.5 months. Among the cohort, 13 children were diagnosed with electrical status epilepticus during slow wave sleep (ESES) syndrome, while 8 had another type of early-onset epileptic encephalopathy (EE), including 6 cases of Lennox-Gastaut syndrome and 2 cases of infantile spasms syndrome. There were no instances of premature discontinuation of IVMP therapy among the patients. Additionally, all our patients with structural epilepsy were deemed ineligible candidates for epilepsy surgery.

Table 1 displays the outcome data observed during 9 months of IVMP pulse therapy and the subsequent 6 months post-therapy.

### 3.2. Seizure Outcome

At the conclusion of IVMP therapy period (9 months), the responder rate, defined as a reduction in seizure frequency of at least 50%, stood at 85.7%. Within this group, 52.4% of patients achieved complete seizure freedom, 9.5% were classified as good responders, and 23.8% were categorized as partial responders (Table 2).

During the initial 6 months of IVMP therapy, the fourth month emerged as the period wherein at least a partial response was observed in more than 50% of the population, with a responder rate of 76%. This included 14.3% of seizure-free patients, 19% classified as good responders, and 42.7% as partial responders. This finding suggests that the fourth month represents the earliest time point for evaluating the efficacy of the therapy (Table 2).

Figure 1A illustrates that among patients who did not achieve a response in the fourth month of IVMP therapy, 60% continued to exhibit no response in the sixth and ninth months of treatment, maintaining this status at the last visit (6 months of follow-up). Conversely, the remaining 40% of subjects who did not demonstrate a response at the fourth month of IVMP therapy displayed a partial response in the sixth and ninth months of treatment, maintaining a favorable outcome at the last visit, albeit with a responder rate of less than 75%.

Figure 1B shows that all patients who were seizure-free at 4 months of IVMP therapy (14.3%) maintained this status until the last visit (6 months of follow-up). Additionally, Figure 1C illustrates that all patients who exhibited at least one partial response after 4 months of IVMP therapy showed a consistent trend of improvement in the subsequent months of therapy, sustaining this trend until the last visit.

The results obtained at 6 months of IVMP therapy and upon completion of the therapy mirrored those seen at the 6-month follow-up, with a favorable outcome evident in 80.9% of patients, including 57.1% who achieved seizure freedom (Figure 1D).

Table 1 presents the association between the outcome and the variables analyzed in our study. A significant association was identified between the outcome and the variables’ etiology (*p* = 0.0475), indicating a better outcome in genetic epilepsy compared to structural epilepsy, and epilepsy type (*p* = 0.0475), suggesting a better outcome observed in ESES compared to other early-onset EE types. These associations were determined using Fisher’s exact test and Wilcoxon rank-sum tests. However, multiple logistic regression analysis did not identify etiology and epilepsy type as the main predictor of response to IVMP therapy at the 6-month follow-up. For etiology, the odds ratio (OR) was 5.2 with a 95% confidence interval (CI) of 0.3–86.9, while for epilepsy type, the OR was 0.2 with a 95% CI of 0.01–3.2. At the last follow-up visit, only one patient with ESES syndrome experienced a relapse (4.7%) after a partial response that had been stable from the fourth to ninth months of therapy.

### 3.3. QOL and Postural-Motor Outcome

At the last follow-up visit, 100% of patients experienced an improvement in quality of life. Among these patients, 36.8% reported improvement after the very first month of IVMP therapy, 21% after 2 months, 5.3% after 3 months, 31.6% after 4 months and 5.3% after 5 months of IVMP therapy.

These results were consistent with the improvement of background activity, appearing more organized, and with a reduction of interictal epileptiform abnormalities observed during follow-up EEGs.

No adverse events were reported, except for a slight irritability observed in 15% of cases during the days of the intravenous infusion.

Finally, an improvement in postural–motor neurological examination was observed in 76.2% of the total sample.

## 4. Discussion

Treatment strategies for epileptic encephalopathies (EE) encompass a variety of approaches, including anti-seizure medications (ASMs), ketogenic diet, vagus nerve stimulation, steroids/adrenocorticotropin hormone therapy, and targeted therapies [4,5,6,7,8,9].

Steroids, such as prednisolone, prednisone, ACTH, methylprednisolone, and hydrocortisone have been employed in the treatment of various seizure disorders, including epileptic spasms [42,43,44,45,46,47], epilepsy syndromes with spike-and-wave activity in sleep [13,48,49,50,51,52], Lennox-Gastaut syndrome [10,23,27,53,54], drug-resistant epilepsies [25,55,56,57,58,59], and status epilepticus [60,61].

Steroid treatment is grounded in the recognition of inflammatory processes in epileptogenesis, suggesting a bidirectional association between seizures and the inflammatory process [62,63,64,65].

Although the precise mechanism by which steroids modulate seizure frequency remains elusive, several hypotheses have been proposed. One of the most widely accepted hypotheses involves the interaction of steroids with the γ-aminobutyric acid (GABA) receptor. This interaction prolongs the duration and frequency of the ligand-gated chloride channel opening, thereby suppressing potential hyperexcitability [62,63,64,65].

Furthermore, it is well established that stress is a common seizure trigger in patients with epilepsy [66,67,68]. Seizure susceptibility and accelerated epileptogenesis associated with stress, particularly chronic stress, are thought to be mediated by stress hormones through their proconvulsant actions [69,70,71]. Moreover, exogenous stress hormones such as corticosterone and corticotropin-releasing hormone (CRH) have been shown to exert proconvulsant actions similar to those of chronic stress [72]. Interestingly, basal levels of corticosterone are elevated in patients with epilepsy and are further increased following seizures, suggesting that the regulation of the hypothalamic–pituitary–adrenal (HPA) axis may be fundamentally altered in epileptic patients [73,74,75,76].

Based on these observations from the literature, it is conceivable that pulse therapy with methylprednisolone, in addition to exploiting the anti-seizure effect linked to the anti-inflammatory action of the pulse phase, acts by reducing hyperexcitability through blocking the HPA axis during the inter-pulse phase. Following this rationale, which involves blocking the HPA axis, we refrain from gradually tapering off steroids via oral cortisone after pulse therapy.

Usually, corticosteroids are typically well-tolerated, but moderate to severe short- and long-term adverse effects can occur [77,78,79,80,81].

In our study, no adverse events were observed, except for slight irritability reported in 15% of cases during the infusion period.

Because of the excellent safety profile and the efficacy of pulse intravenous methylprednisolone, it is frequently employed in patients with drug-resistant epilepsy, including epileptic encephalopathies. However, different therapeutic strategies are reported, in terms of dosage (15–30 mg/kg/day), frequency (1 pulse per week/1 pulse per month), and duration (1–36 months) [19,20,21,22,23,24,25,26,27,28].

In our study, we administered 20 mg/kg/day of intravenous methylprednisolone over three consecutive days [13]. Multiple cycles of IVMP were administered at monthly intervals for a total period of 9 months.

Our population analysis revealed that 61.1% of patients achieved seizure freedom, 11.1% achieved more than 75% seizure reduction, and 27.8% experienced between 50% and 75% seizure reduction. These results were comparable with those of studies in the literature, although previous studies, mainly retrospective, have reported improvement with steroids ranging from 40% to 73% [19,20,21,22,23,24,25,26,27]. The wide variation in response could be due to the different populations studied and the use of different types of preparations, routes and duration of steroid administration [19,20,21,22,23,24,25,26,27].

In the literature, there is only one randomized controlled study that analyzed 80 children with EE undergoing IVMP pulse therapy. In this study, therapy was continued for 3 months, showing significant improvement in seizure frequency, EEG parameters, and adaptive functioning, without serious steroid-related adverse effects. After 4 months of follow-up, 75% of patients treated with IVMP had a seizure reduction > 50% compared to 15.4% in the control group [28].

At the conclusion of the IVMP therapy period (9 months), the outcomes achieved were sustained until the last visit (6-months follow-up), with a favorable outcome observed in 80.9% of patients, including 70.5% who were seizure-free.

Careful analysis of our results reveals that the fourth month of IVMP therapy was the first time point during the treatment where we could outline a reliable outcome trajectory.

Indeed, all patients who were seizure-free in the fourth month remained seizure-free until the sixth month of follow-up and patients who did not achieve a response in the fourth month of IVMP therapy demonstrated either no response (60%) or a responder rate < 75% (40%) (Figure 1A,B).

Additionally, the subjects classified as “good responders” demonstrated an improving trend until achieving seizure-free status at the conclusion of follow-up. Finally, the “partial responders” showed a stable or improving trend in 88.9% of subjects (Figure 1C).

Furthermore, upon analyzing our results, it emerged that the sixth month of therapy was the first time point practically showing comparable data observed at the 6- month follow-up (Figure 1D).

Genetic epilepsy and ESES were found to be significant predictors of a favorable clinical outcome (Table 1). While we cannot determine if these two variables indicate severity, based on seizure frequency, it appears that non-responders exhibited a more severe form of epilepsy. To perform a logistic regression with these data, the two significant variables, namely etiology and epilepsy type, should be included in the regression model. However, upon conducting the regression, the association does not remain significant.

In epileptic encephalopathies, unlike other forms of epilepsy, the treatment goal encompasses not only seizure control but also the prevention of further deterioration in cortical network functioning. Unfortunately, achieving this objective in epileptic encephalopathies solely with anti-seizure medications is challenging. Indeed, the most effective therapies for obtaining these outcomes include ketogenic diet, vagus nerve stimulation (VNS), and steroid therapy [7,8,9].

In our study, 100% of patients demonstrated an improvement in QoL at the conclusion of the follow-up period, with improvements observable as early as the first month of therapy (36.8%). Furthermore, the improvements in QoL were perfectly consistent with the enhancements observed in postural–motor development and in EEG features, such as background activity and interictal abnormalities, throughout treatment and at the last visit of follow-up.

As we previously suggested regarding VNS therapy, steroid therapy should also be initiated as early as possible in patients with epileptic encephalopathy to prevent the encephalopathic effects of epilepsy and potentially the establishment of aberrant circuits during a critical period of brain maturation [7].

This concept would likely result in more pronounced effects in very young children, who exhibit greater brain plasticity, and for epileptic encephalopathies, as defined by stating that “the epileptic activity itself may contribute to severe cognitive and behavioral impairments above and beyond what might be expected from the underlying pathology alone (e.g., cortical malformation), and that these can worsen over time” [2,7].

It is well known that the organization of brain networks is based on small-worldness and modularity, and this organization is altered in neurological disorders, including epilepsy [82,83,84,85,86,87,88]. It has also been reported that in epilepsy, the functional topology of the brain network is altered, which increases vulnerability to seizures [87,88,89,90,91,92,93,94,95]. These observations are consistent with the theory that epilepsy is a network disorder, and seizures occur due to the anomalous topology of structural and functional networks [96,97,98].

We speculate that steroid therapy induces integrated network organization, a more balanced topology, and less pathological architecture, leading to a more efficient reorganization of functional brain networks and network structure, potentially contributing to the clinical improvement observed in our population. Based on these observations, we speculate that early intervention leads to a widespread reorganization of brain networks and prevents the establishment of aberrant circuits associated with the encephalopathic state. This may influence complex processes underlying drug resistance in epileptic encephalopathies. The hypothesized large-scale reorganization of brain networks may also explain why IVMP therapy not only affects seizure frequency but also modifies quality of life. The overall clinical improvement founded in our patients with EE was accompanied by improved caregiver QoL. This outcome potentially reduces caregiver burnout and fosters a more positive patient–caregiver interaction.

Therefore, our data suggest that IVMP therapy should be considered as early as possible in the management of epileptic encephalopathies.

Our clinical findings are supported by the improved background and interictal activity observed on EEGs following IVMP therapy. The improvement of EEG background activity reflects the amelioration of cerebral electrogenesis, which is an indirect indicator of maturation and/or reorganization of functional networks. This improvement, as we speculate, is favored by IVMP pulse therapy.

Without studies that carefully analyze changes in EEG background activity after IVMP pulse therapy, we cannot exclude the possibility that EEG background features could improve through brain maturation alone, particularly in early life. Only further investigation into the direct effects of early IVMP pulse therapy on EEG will resolve this issue.

Similarly, the postural–motor improvement found in our patients is likely the result of the improvement in the background activity found on the EEG and therefore of the maturation/reorganization of the functional networks.

Finally, our study confirms the high tolerability of IVMP pulse therapy, with no dropouts observed.

The most significant limitation of our study is its retrospective methodology. However, retrospective studies offer the advantage that results are not predetermined, as all evaluations are based on existing data sources where both exposure and outcomes are readily available. Moreover, another important limitation of our study is the absence of a control group. Nevertheless, this bias is partially mitigated when considering the syndromic conditions of our population (ESES syndrome, Lennox-Gastaut syndrome and infantile spasms syndrome). These conditions are epileptic encephalopathies characterized by, for definition, severe drug resistance and serious clinical manifestations, in which the epileptic activity itself can contribute to serious cognitive and behavioral disorders that go beyond what could be expected from the underlying pathology alone [2,3].

Finally, our sample size was small and heterogeneous, preventing us from considering our results conclusive. Therefore, further studies with adequate population sizes are necessary to validate our findings.

## 5. Conclusions

In conclusion, our study confirms the efficacy and high tolerability of IVMP pulse therapy in pediatric patients with epileptic encephalopathies. We observed improvements in quality of life, EEG features, and postural–motor development. Our data suggest that IVMP pulse therapy should be considered promptly in epileptic encephalopathies, with the fourth month of treatment identified as a critical time point for correctively assessing therapeutic efficacy. Furthermore, genetic etiology and ESES emerged as potential significant predictors of a favorable clinical outcome.

## Figures and Tables

**Figure 1 jcm-13-02497-f001:**
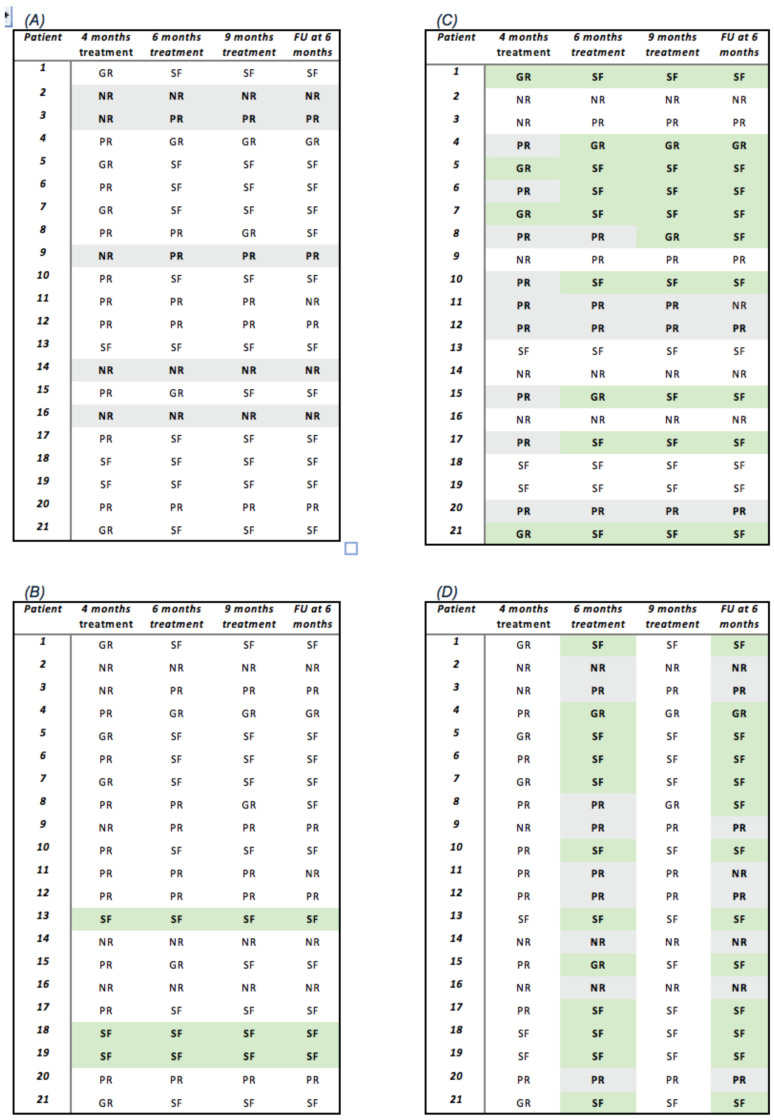
We decided to highlight in different colors the clinical outcome of our patients at different time point: in grey non responders (NR) and partial responders (PR), and in green good responders (GR) and seizure-free (SF) patients. (**A**) shows that patients who were NR in the fourth month of IVMP therapy remained NR (60%) or became PR (40%); (**B**) shows that patients who were SF in the fourth month remained SF until the sixth month of follow-up; (**C**) shows that 100% of patients who were GR at fourth month of IVMP therapy demonstrated an improving trend becoming SF at the end of follow-up, and patient who were PR showed a stable or improving trend except for 1 patient who became NR; (**D**) showed that the sixth month of therapy was the first time point practically showing comparable data observed at the 6-month follow-up. Legend: NR: non responders; PR: partial responders; GR: good responders; SF: seizure-free; FU: follow-up.

**Table 1 jcm-13-02497-t001:** Outcome data during 9 months of IVMP pulse therapy and at 6 months of follow-up posttreatment.

	4 Months Treatment			6–9 Months Treatment			6 Months Posttreatment		
	Responders	Non- Responders		Responders	Non- Responders		Responders	Non- Responders	
	N = 16	N = 5	*p*-Value	N = 17	N = 4	*p*-Value	N = 16	N = 5	*p*-Value
	76.2%	23.8%		81%	19%		76.2%	23.8%	
Male gender N (%)	9 (56.3)	3 (60)	0.9999	10 (58.8)	2 (50)	0.9999	10 (62.5)	2 (40)	0.6108
Age at epilepsy onset (mo)									
Mean (SD)	28.8 (25.9)	25 (42.7)	0.3412	27.9 (25.3)	28 (48.7)	0.2618	26.9 (25.9)	31 (42.7)	0.5079
Median (range)	17.5 (6.5)	4 (4–13)		13 (7–43)	4 (3.552.5)		12.5 (6.5–51)	4 (4–43)	
Age at treatment beginning (mo)									
Mean (SD)	94.2 (39.1)	133.4 (76.3)	0.3855	93.1 (38.2)	147.8 (80)	0.2094	95.4 (38.2)	129.6 (80.3)	0.6794
Median (range)	85.5 (71.5–121.5)	107 (76–156)		82 (72–117)	131.5 (89.5–206)		85.5 (74–121.5)	107 (72–156)	
Epilepsy duration pretreatment (mo)									
Mean (SD)	65.4 (34.6)	108.4 (44.1)	0.1265	65.2 (33.5)	119.8 (41.7)	0.0438	68.4 (31.8)	98.6 (59.5)	0.3018
Median (range)	69.5 (39.5–82)	104 (68–152)		69 (42–79)	128 (86–153.5)		69.5 (49–82)	104 (68–152)	
Epilepsy etiology N (%)									
Structural	4 (25)	4 (80)	0.0475	4 (23.5)	4 (100)	0.0117	4 (25)	4 (80)	0.0475
Genetic	12 (75)	1 (20)		13 (76.5)	0 (0)		12 (75)	1 (20)	
Epilepsy syndrome N (%)									
ESES	13 (81.3)	0 (0)	0.0028	13 (76.5)	0 (0)	0.0117	12 (75)	1 (20)	0.0475
Other early onset EE	3 (18.7)	5 (100)		4 (23.5)	4 (100)		4 (25)	4 (80)	
Seizure frequency pretreatment N (%)									
Multi-daily	7 (43.8)	4 (80)		8 (47.1)	3 (75)		7 (43.8)	4 (80)	
Multi-weekly	1 (6.2)	1 (20)	0.3054	1 (5.9)	1 (25)	0.4762	1 (6.3)	1 (20)	0.3054
Multi-monthly	6 (37.5)	0 (0)		6 (35.3)	0 (0)		6 (37.5)	0 (0)	
Multi-yearly	2 (12.5)	0 (0)		2 (11.7)	0 (0)		2 (12.5)	0 (0)	

Legend: N: number; mo: months; SD: standard deviation; ESES: encephalopathy related to status epilepticus during slow sleep; EE: epileptic encephalopathies.

**Table 2 jcm-13-02497-t002:** IVMP pulse therapy efficacy and outcome.

Outcome	4 Months Treatment N (%)	6 Months Treatment N (%)	9 Months Treatment N (%)	6 Months Posttreatment N (%)
**Seizure-free**	3 (14.3)	10 (47.6)	11 (52.4)	12 (57.1)
**Good responder**	4 (19)	2 (9.5)	2 (9.5)	1 (4.8)
**Partial responder**	9 (42.7)	6 (28.6)	5 (23.8)	4 (19)
**Non-responders**	5 (24)	3 (14.3)	3 (14.3)	4 (19.1)

**Legend**: Good responders: >75% seizure reduction; partial responders: 50–75% seizure reduction.

## Data Availability

The data presented in this study are available on request from the corresponding author.

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
