# Peer review of "Efficacy and Safety of Pulse Intravenous Methylprednisolone in Pediatric Epileptic Encephalopathies: Timing and Networks Consideration"

_jcm, 2024, doi:10.3390/jcm13092497_

Round 1

Reviewer 1 Report

Comments and Suggestions for Authors

The authors of this study conducted a retrospective analysis on the efficacy and safety of pulse intravenous methylprednisolone in pediatric epileptic encephalopathies. The title of this study is worthy of investigation. However, upon reviewing the manuscript, several points are highlighted below that require further clarification, explanation, or modification:

1.       Is there documentation of ethics approval from a recognized institution? Kindly provide the ethics approval number. Were patients or their legal guardians required to complete consent forms?

2.       Were the patients receiving anti-epileptic therapy prior to or during the study? If so, what steps were taken to minimize bias relating to variations in medications and treatment durations? It is crucial to consider the potential synergistic effects of corticosteroids with existing anti-epileptic medications.

3.       Has information about the patients' underlying conditions been disclosed? How was bias mitigated concerning this aspect?

4.       Was a control group included in the study? If not, how were the results compared to scenarios where patients did not receive any medication or only underwent conventional anti-epileptic therapy? Please provide clarification.

5.       In reference to line 193, how was quality of life assessed? What criteria and parameters were used to gauge improvement? If feasible, consider providing the questionnaire utilized for this purpose as a supplementary document.

6.       Can you define what constitutes improvement in EEG findings? Please provide an evidence-based elaboration.

7.       Was the sample size adequate for the study? Kindly elucidate on the efficiency of the sample size.

8.       Please cite and reference relevant studies similar to yours, outlining their methodologies and findings.

9.       Highlight the originality and novelty of your study in comparison to existing literature.

10.   Summarize the practical and policy implications arising from your study results.

11.   Incorporate references from 2023 in your study, exploring recent efforts and advancements in this field encountered over the past year.

Wishing you success and prosperity in the preparation of your manuscript.

Comments on the Quality of English Language

Minor editing of English language required.

Reviewer 2 Report

Comments and Suggestions for Authors

Dear Authors,

You are kind to present data on a topic of drug-resistant epilepsy and epileptic encephalopathy, which is a very serious problem in epileptology. You are presenting a very good outcome with over half of the patients reaching freedom from seizure. Study with such good results should be well described, therefore I have some comments/suggestions

1. Was all the patients consulted or assessed as potential candidates for epileptic neurosurgery and disqualified from such a therapy?

2. Table 1 showing effects of the therapy should be edited. It is  confusing in its actual form. 

3. You were giving steroids for 9 months. Why just 9 months? Any literature data or your personal experience showing (for instance) that 9 months is better than 6 months and also better than 12 months ? 

4. You have written that EEG data were also collected. Please, provide data if clinical improvement was paralleled with changes in EEG?

5. Please, provide some clinical data about those 21 patients. What type of epilepsy/ epileptic syndrome/ type of seizure did they present? Did they have any serious neurodevelopment problems? If so - what kind of? How were they selected of corticosteroid therapy? 

Round 2

Reviewer 1 Report

Comments and Suggestions for Authors

There are some fundamental errors in your study design, specifically the lack of a control group and a small sample size, which renders your results unreliable. Moreover, some of the errors that have been identified have not yet been corrected.

Comments on the Quality of English Language

Minor English language editing was recommended in previous version of the manuscript that has not been applied yet. 

Reviewer 2 Report

Comments and Suggestions for Authors

Dear Authors,

Thank you for correcting your manuscript. 

Author Response

Dear Reviewer, thank you for helping us improve our study.

All the best